# Development of Flame-Retardant Polylactic Acid Formulations for Additive Manufacturing

**DOI:** 10.3390/polym16081030

**Published:** 2024-04-10

**Authors:** Robert Aguirresarobe, Itxaso Calafel, Sara Villanueva, Alberto Sanchez, Amaia Agirre, Itxaro Sukia, Aritz Esnaola, Ainara Saralegi

**Affiliations:** 1POLYMAT and Department of Advanced Polymers and Materials: Physics, Chemistry and Technology, Faculty of Chemistry, Universidad del País Vasco/Euskal Herriko Unibertsitatea, UPV/EHU, 20018 San Sebastian, Spain; roberto.hernandez@ehu.eus (R.A.); itxaso.calafel@ehu.eus (I.C.); amai.agirre@ehu.eus (A.A.); 2TECNALIA, Basque Research and Technology Alliance (BRTA), Parque Tecnológico de San Sebastián, 20009 San Sebastian, Spain; sara.villanueva@tecnalia.com; 3Department of Mechanics and Industrial Production, Mondragon Unibertsitatea, 20500 Arrasate-Mondragon, Spain; isukia@mondragon.edu (I.S.); aesnaola@mondragon.edu (A.E.); 4Group ‘Materials + Technologies’, Department of Chemical and Environmental Engineering, Faculty of Engineering of Gipuzkoa, Universidad del País Vasco/Euskal Herriko Unibertsitatea, UPV/EHU, 20018 San Sebastian, Spain

**Keywords:** poly(lactic acid), flame retardants, additive manufacturing, railway standard, mechanical properties, processability

## Abstract

Polymeric materials, renowned for their lightweight attributes and design adaptability, play a pivotal role in augmenting fuel efficiency and cost-effectiveness in railway vehicle development. The tailored formulation of compounds, specifically designed for additive manufacturing, holds significant promise in expanding the use of these materials. This study centers on poly(lactic acid) (PLA), a natural-based biodegradable polymeric material incorporating diverse halogen-free flame retardants (FRs). Our investigation scrutinizes the printability and fire performance of these formulations, aligning with the European railway standard EN 45545-2. The findings underscore that FR in the condensed phase, including ammonium polyphosphate (APP), expandable graphite (EG), and intumescent systems, exhibit superior fire performance. Notably, FR-inducing hydrolytic degradation, such as aluminum hydroxide (ATH) or EG, reduces polymer molecular weight, significantly impacting PLA’s mechanical performance. Achieving a delicate balance between fire resistance and mechanical properties, formulations with APP as the flame retardant emerge as optimal. This research contributes to understanding the fire performance and printability of 3D-printed PLA compounds, offering vital insights for the rail industry’s adoption of polymeric materials.

## 1. Introduction

Additive manufacturing (AM), a cutting-edge production technology, exhibits the potential to diminish stockpiles by enabling on-demand part fabrication, thereby offering significant prospects for optimizing operational efficiency. This technology has increasingly captivated, among others, the railway industry. Major players like the French National Railway System (SNCF), Alstom, MGA, Bombardier, CAF, and SIEMENS are actively exploring and embracing additive manufacturing technologies to revolutionize spare polymer-based parts production [1]. However, a persistent challenge hindering polymer-based AM applications, particularly in safety-critical fields such as the railway industry, is their inherent flammability. Applications such as interior panels, seating, cable insulation, and housing for electronic equipment demand novel strategies to mitigate these risks effectively [2,3]. This concern is exacerbated by the porous nature of 3D-printed components, necessitating innovative approaches to meet stringent fire safety standards.

To address this challenge, flame retardant additives have been commonly introduced into polymers, demonstrating effectiveness across a wide range of polymers and applications [4,5]. However, polymer hot melt extrusion (HME)-based additive manufacturing (AM) demands precise material rheology for successful printing, posing challenges in incorporating additives with specific functionalities such as flame retardants [6,7,8]. Therefore, solutions that simultaneously meet fabrication requirements and fulfill fire retardancy criteria remain largely unexplored.

This study addresses this gap by focusing on the gold standard polymer matrix, poly(lactic acid) (PLA), and employing a variety of commercially available halogen-free flame retardants. PLA exhibits favorable properties, such as biodegradability and ease of processing. Moreover, PLA is environmentally friendly and derived from renewable resources, making it an attractive choice for various applications, including additive manufacturing [9,10]. However, it is important to acknowledge that PLA also has several disadvantages, including weak resistance to UV, low-glass transition temperature, etc. These limitations may restrict its suitability for certain applications and warrant careful consideration when selecting PLA as a material for specific projects or industries. Furthermore, PLA has a relatively low melting temperature, typically around 150–160 °C, which facilitates its processing in extrusion-based AM systems [11]. Thus, hot melt extrusion has emerged as a prominent technique for processing PLA polymers in additive manufacturing. The literature reports have highlighted the efficacy of HME in achieving a uniform distribution of additives and enhancing the mechanical properties of PLA-based materials [12,13,14]. This process involves melting the PLA polymer at elevated temperatures, mixing it with additives, and extruding it through a die to form filament or pellets suitable for 3D printing. PLA has found widespread utilization in the fields of drug delivery [15], food packaging [16], the automotive industry [8], railway [17], etc. These applications highlight the versatility of PLA in various industries, further underscoring its significance in additive manufacturing.

Previous research has investigated the performance of flame retardants in PLA, encompassing blends of conventional flame retardants like ammonium polyphosphate (APP) and expandable graphite (EG) [18], the impact of nanofillers combined with flame retardants (such as nanoclays with APP [19], aluminum hydroxide (ATH) or EG) [20,21,22], various intumescent flame retardant systems involving APP, charring agents (e.g., pentaerythritol), blowing agents like tris (2-hydroxyethyl) isocyanurate (THEIC) [23] or melamine [24], and biobased additives, such as lignin [25]. Additionally, more complex structures, including spirocyclic pentaerythritol bisphosphorate disphosphoryl melamine (SPDPM) [26], phosphorus–nitrogen flame retardants (PNFRs) [27], chitosan-based formulations [28], and hyperbranched charring agents with APP [29], have been explored. These flame-retardant systems enhance the fire performance of PLA in diverse ways.

Recent studies have addressed the use of flame retardants on 3D-printed polymeric composites [30]. One study investigated intumescent formulations for PLA composed of ammonium polyphosphate, lignin, and acidic-activated montmorillonites, which influence fire performance based on the acidity rate [31]. Another study demonstrated that a mere 2 wt% of APP and 0.12 wt% of resorcinol bis(diphenyl phosphate) (RDP) reduce the flammability of PLA, enabling the composite to easily attain the UL94 V-0 rating, with RDP acting as a compatibilizer of APP and PLA [32]. Additionally, research is also focused on the addition of melamine polyphosphate (MPP) as a flame retardant for PLA and its application in 3D printing. Their findings revealed that the addition of MPP renders the PLA matrix brittle; however, the presence of nanoclays restores the impact strength, with both additives affecting the rheology of the polymer blend [33]. Despite the significant interest in this area, there remains a paucity of comprehensive studies examining the impact of flame retardants on the processability of PLA polymers: a crucial step toward the development of printable flame-retardant PLA grades.

Hence, the main goal of this work was to establish a material selection protocol in terms of fire performance and 3D printability, facilitating the streamlined identification of materials for flame-retardant 3D printing. A comprehensive analysis encompasses fire retardancy, the influence on thermal and mechanical properties, printability, and final object performance. Moreover, rigorous fire tests adhere to European railway regulations [34], ensuring the practical relevance and compliance of the developed flame-retardant PLA formulations in real-world applications, particularly in industries like railways where both safety and efficiency are paramount.

In addition to addressing the technical aspects of our study, it is crucial to provide a clear justification for our research objectives. The railway industry presents an ideal scenario for the application of flame-retardant PLA due to stringent safety regulations and the need for materials that can withstand high temperatures and reduce fire hazards. By incorporating flame-retardant properties into PLA, we aim to address safety concerns associated with traditional materials while taking advantage of PLA’s inherent benefits, such as its better environmental profile, biodegradability, and ease of processing. This aligns with sustainability initiatives aimed at reducing environmental impact and enhancing passenger safety.

## 2. Materials and Methods

### 2.1. Materials

Poly(lactic acid) (PLA) Ingeo™ 3D450 NatureWorks (Minnetonka, Minneapolis, MN, USA), with a molecular weight of 150,300 g·mol^−1^, is used as the polymer matrix. Different commercial halogen free-flame retardants (FRs) were incorporated as follows: Budit^®^ 620, melamine-coated ammonium polyphosphate, APP, from Budenheim (Budenheim, Germany), expandable graphite Firecarb TEG 345, EG, from LKAB Minerals (Luleå, Sweden), aluminum hydroxide Apyral 120, ATH, from Nabaltec (Schwandorf, Germany), cresyl diphenyl phosphate (CDP) Disflamoll^®^, DPK, from Lanxess (Cologne, Germany) and an intumescent flame retardant, IM, prepared in situ by the combination of the following additives: ammonium polyphosphate, Exolit^®^ AP 422, from Clariant (Muttenz, Switzerland), melamine, MEL, Melafine^®^ from Oci Melamine (Barcelona, Spain) and a polyhydric alcohol, PA, Charmor™ PM40 from Perstorp (Malmö, Sweden). Moreover, Irganox B215 was used as a thermal stabilizer.

### 2.2. Preparation of Samples

PLA and flame retardants were dried at 50 °C for 8 h before every sample preparation. Then, several formulations based on the PLA matrix and containing different flame retardants were prepared by melt blending in an internal mixer, Plastograph^®^ EC from Brabender. The mixing was performed through two steps at the same temperature (180 °C) and rotor speed (30 rpm) conditions: in the first step, the PLA and the thermal stabilizer, Irganox B215, were added until the polymer melted (6 min). In the second step, the flame retardant was added to achieve a well-dispersed mixture. The total mixing time was 15 min. The batch was extracted from the mixing chamber manually and then cooled under air until reaching room temperature. The obtained samples were ground in a cutting mill SM 300 from Retsch at 700 rpm using a 6 mm mesh. Specimen type 1A (1 mm thickness) for tensile characterization and specimens for flexion characterization (80 mm × 10 mm × 4 mm) were prepared by 3D printing and the injection process. The composition percentages of the materials mentioned in Table 1 were selected based on a combination of factors, including compatibility with PLA, desired fire-resistance properties, and the consideration of mechanical performance.

Regarding 3D-printed samples, they were prepared in a Delta Wasp 4070 3D impression machine with a pellet extruder. The printing volume was 400 × 400 × 700 mm^3^, and a climatized and closed chamber was used to allow heat all around. The printer utilized a nozzle with a diameter of 0.8 mm, facilitating the precise deposition of material. The infill pattern was set vertically, enhancing the structural integrity of the printed object. Maintaining a printing temperature of 185 °C ensured the proper flow and adhesion of the filament, while the base temperature was set at 40 °C to promote adhesion to the build platform. A printing speed of 60 mm/s was employed to strike a balance between efficiency and print quality. The layer height was set at 0.3 mm, contributing to the overall resolution and detail of the printed layers.

On the other hand, in the case of injected samples, specimens were injection-molded under controlled conditions using a constant mold temperature of 60 °C. The injection process involved a fixed injection time of 10 s and a variable injection volume ranging from 20 to 50 cm^3^/s, with a consistent pressure of 150 bar applied throughout. Additionally, a variable temperature profile was employed during the molding process, including temperatures above 160 °C.

Finally, specimens for thermo-mechanical (25 mm × 5 mm × 1.5 mm) and cone calorimetry (100 mm × 100 mm × 3 mm) evaluations were obtained by compression molding using an LP-S-50 (LabTech, Sorisole, Italy) hydraulic press as well as 3D printing. Samples were preheated at 190–195 °C without pressure for 3–4 min, and thereafter, a pressure cycle of 70 kN was applied for 1 min.

### 2.3. Thermal, Rheological, Mechanical and Fire Behavior

The thermal stability of the samples was analyzed by thermogravimetric analysis (TGA) using a TGA Q500 thermal analyzer from TA Instruments (New Castle, DE, USA). Samples were heated from room temperature to 600 °C at a heating rate of 10 °C/min under a nitrogen atmosphere.

The molecular weights and number of average molar masses, M_w_ and M_n_, respectively, of the samples were determined by gel permeation chromatography (GPC) using a Waters 717 Autosampler chromatograph (Milford, MA, USA), consisting of a pump, a refractive index detector, and Waters Styragel (HR2, HR4, and HR6) columns. The analysis was carried out at 35 °C using tetrahydrofuran (THF) as an eluent (flow rate of 1 mL/min). Measured distributions were referred to polystyrene narrow standards ranging from 580 to 395 · 10^3^ g/mole and corrected with the universal calibration using the Mark Houwink parameters for polystyrene: K = 1.58 × 10^4^ mL/g, α = 0.704.

The crystallization and melting temperatures of the polymers were measured by Differential Scanning Calorimetry (DSC) with a TA DSC25 equipped with an Intracooler. Ultra-pure nitrogen was used as a purge gas. Samples with 7–8 mg of approximate weight were encapsulated in aluminum pans. Tin and indium standards were used as calibrates. Samples were heated from 0 °C to 100 °C at a heating rate of 20 °C/min; then, they were held at 200 °C to erase thermal history. Later, they were cooled to 0 °C at 20 °C/min. After 2 min of equilibration at 0 °C, a second heating scan was recorded between 0 and 200 °C. The degree of crystallinity, X_c_, was calculated as follows:(1)$$X_{c}=\frac{\Delta H_{m}-\Delta H_{\mathrm{cc}}}{\Delta H_{m}^{{}^{\circ}}\;\cdot \;\left(1-\alpha \right)}\;\cdot \;100$$
where ΔH_m_ (J/g) is the experimentally obtained melting enthalpy of the sample, ΔH_cc_ (J/g) is the cold crystallization enthalpy, $\Delta H_{m}^{{}^{\circ}}\;$ is the equilibrium melting enthalpy (a value of $\Delta H_{m}^{{}^{\circ}}$ = 93 J/g for neat PLA was employed, as reported in the literature [35]) and α is the amount of filler, in this case, FR.

The dynamic mechanical behavior of the samples was analyzed by dynamic mechanical analysis (DMA). To this end, samples were cut into strips of 25 mm × 5 mm × 1.5 mm (length × width × thickness) and were tested in tensile mode on an Eplexor Gabo 100N analyzer from Netzsch, using a static strain of 0.10%. The temperature varied from −100 to 150 °C at a scanning rate of 2 °C/min and a fixed operation frequency of 10 Hz.

A fire performance evaluation was carried out by cone calorimetry in FTT equipment according to ISO 5660 [36] under a heat flux of 50 kW/m^2^ for 1200 s. The tested sample dimensions were 100 mm × 100 mm × 3 mm. The time to ignition (TTI, s), peak Heat Release Rate (HRRpeak, kW/m^2^), Total Heat Release (THR, MJ/m^2^), Maximum Average Rate Heat Emission (MARHE, kW/m^2^), Total Smoke Production (TSP, m^2^), Smoke Extinction Area (SEA, m^2^/kg) and CO/CO_2_ ratio were recorded. Two repetitions of each sample were performed. The sample holder was covered with metal mesh to prevent the deformation and loss of material when the intumescence of samples occurs.

The melt viscosity at processing temperatures of the samples was characterized by extrusion flow experiments, which were performed in a Göttfert Rheograph 25 rheometer using a capillary die with L/D = 30/1. Small amplitude oscillatory shear (SAOS) experiments were conducted in linear viscoelastic conditions in order to obtain the viscoelastic behavior of the samples at different temperatures. These experiments were carried out in an ARES rheometer (TA Instruments), using a 25 mm parallel plate geometry.

### 2.4. Mechanical Characterization of the Printed and Injection-Molded Samples

Tensile and flexural measurements were performed using a universal testing machine. Tensile tests were performed in accordance with ISO 178 [37], and each specimen was tested to failure at 23 °C at a crosshead speed of 1 mm/min. Flexural strength and modulus tests were performed according to ISO 527 [38]. The support span length was 5 cm. The head speed was 1 mm/min. Test specimens for tensile and flexural measurements were prepared both by the injection and 3D printing processes.

## 3. Results and Discussion

This study pretends to establish a criterion to be applied to the selection of 3D printable fire-retardant PLA. These materials should fulfill different requirements in terms of functionality, printability and mechanical performance. In Table 1, the compositions of the studied samples are summarized, specifying the type and amount of flame retardant added in each case. Selected compositions have been defined according to the range of effectiveness of these flame retardants in view of the information available in the literature. Flame retardants acting mainly in the condensed phase APP, EG, IM, and on the gas phases ATH and CDP were selected.

### 3.1. Fire Behavior

The fire performance of 3D-printed materials is of main relevance for railway applications. Therefore, all prepared mixtures were tested in the cone calorimeter, and the results are shown in Table 2 and Figure 1. In this table, parameters related to heat release (HRR, THR, MARHE) and smoke generation (CO/CO_2_ ratio, SEA) are collected, joined with time to ignition and final residue.

Time is a key factor to guarantee a safe evacuation in the event of fire. ARHE (average rate of heat emission), Figure 1, considers the amount of released heat along combustion, but it is modulated in relation to the time that heat is generated. The MARHE (maximum of ARHRE) (Table 2) is used as the criteria of classification in railway European regulation EN45545 [34]. It is of note that, regardless of the flame retardant used, no self-extinguishable mixtures were obtained within this work. However, in all the cases, MARHE values decreased in relation to the amount of flame retardant presented in the FR-PLA composition, with certain solutions (EG, APP, IM) more effective than others (ATH, CDP). This observed trend aligns with findings reported by other researchers in the scientific community [39,40,41].

This is clearly shown in the Petrella plots, Figure 1B. In this representation, the total heat release (THR) as a function of the HRR peak/ignition time ratio is presented. Thus, an improved flame retardancy is depicted as a value in the left-down part of the plot. Represented data for the different mixtures showed an improvement in fire performance to pure PLA, although with significant differences. Data tend to group in two regions of the graph, and a general trend is observed. As expected, the higher the FR content, the better the fire behavior. However, it is noteworthy that ATH and, especially, CDP-based materials showed poor behavior against fire, even at a high FR content. In contrast, materials containing APP, EG as well as the intumescent mixture showed more effective fire performance.

Figure 2 shows pictures of the obtained residues for cone calorimetric-tested samples. All materials presented a residue, even for reference PLA, corresponding to the inorganic fillers already present in this commercial grade. Between the different samples, better fire performance was directly related to the higher residue remaining after combustion [42]. It is represented in the pictures and correlates with the results presented in Table 2. It can be concluded that the most effective action of flame retardants is produced in the condensed phase, with the special contribution of the intumescent effect of EG, APP, and IM blends. Flame retardants that mainly work in the gas phase, such as CDP, showed minor flame-retardant effects. ATH, acting both in the gas and the condensed phase, has an intermediate performance. This conclusion is consistent with numerous studies reported in the literature. Researchers have consistently observed that flame retardants with strong action in the condensed phase, such as intumescent systems containing EG, APP, and IM, tend to exhibit superior flame-retardant properties. These systems work by forming a protective char layer upon exposure to heat, effectively insulating the underlying material from the flame [43]. Conversely, phosphorous-based flame retardants primarily operating in the gas phase, like CDP, typically offer limited flame-retardant effects as they mainly act by scavenging free radicals and interfering with the combustion process [44]. ATH, known for its ability to release water vapor and cool the material during combustion, occupies an intermediate position due to its dual-phase action [45]. This collective body of evidence underscores the importance of considering both gas and condensed phase mechanisms when evaluating the efficacy of non-halogenated flame retardants in polymeric materials.

Preselected samples (APP, EG, IM) presented a MARHE value below 90 kW/m^2^, which makes them future candidates for use in railway applications according to the EN 45545-2 standard [34]. MARHE values lower than 90 kW/m^2^ compliance with one of the demands for R1 (interior vertical surfaces) and R7 (external surfaces) requirements allowed the use of these materials for certain applications (Appendix A, hazard level 2 (HL2)). PLA with 30% of APP showed a MARHE below 60 KW/m^2^, making it suitable for the most demanding hazard level, HL3 (Appendix A) [34].

The addition of flame retardants to the PLA matrix increased the total smoke production (TSP) as well as the darkness of this smoke (specific extinction area, SEA), except for the expandable graphite and ATH formulations. Released gases interfering in the combustion reaction in the gas phase could explain the higher smoke generation of CDP, APP, and intumescent blend solutions [46]. Additionally, the non-complete combustion induced by flame retardants resulted in an increase in the CO/CO_2_ ratio. Flame retardants able to release phosphorus-based radicals, acting in the gas phase (CDP and in minor proportion APP), clearly showed this effect [47,48].

### 3.2. Effect of FR Addition on the Physico-Chemical, Thermal and Mechanical Properties

It is well-known that the incorporation of flame retardants might interfere with the thermal and thermo-mechanical properties of the polymer matrix, altering the performance of the blend [49]. The temperature and the degradation mechanism of flame retardants and, specifically, the released effluents during the combustion process might interfere with the degradation path of the polymer. The TGA thermograms shown in Figure 3 present the mass change in the samples as a function of temperature. For comparison, each graphic shows the weight loss vs. temperature of reference for PLA, including each individual flame retardant and theoretical and real curve of flame retardant PLA compounds (graphics for the PLA pellet, reference PLA, and Irganox are included in Appendix A). Moreover, the initial mass loss (T_i_) temperature data from the TGA thermograms were obtained considering the loss of 5% of the initial mass, the maximum mass loss rate temperature (T_max_), which was determined by the maximum value of the first derivative; if the mass remained at 600 °C, it was defined as a residue (Table 3).

Reference PLA thermally degrades in a single step, starting at around 330 °C and leaving a residue of 11 wt%. In general, PLA degradation is a complex process involving random chain scission intramolecular transesterifications and selective depolymerization steps highly dependent on composition in terms of stereoisomerism, molecular weight, and the presence of moisture, catalysts, residual monomers, and impurities [50,51]. However, PLA does not leave a residual char even in an oxidative atmosphere. Thus, the measured residue is related to the presence of inorganic fillers on the commercial-grade PLA.

The incorporation of flame retardants significantly affects the thermal stability of the PLA/FR blend since the initial degradation temperature and/or the degradation stages are altered, as is evidenced by the difference between the theoretical (no interaction between PLA and flame retardants) and real performance. Attending to weight loss, the detrimental effect on PLA stability derived from the incorporation of ATH, EG, and IM as flame retardants was noticeable. In these blends, the real curve shows higher weight loss than the theoretical curve. In the case of EG, the significant degradation of PLA/EG mixtures can be produced by the promotion of the hydrolytic degradation of PLA under the action of gases from the decomposition of expandable graphite. This gas, of an acid nature (sulfuric acid) [52], can be released at PLA processing temperatures (170–180 °C). This is an unexpected result, as numerous works have reported the beneficial effect of EG on fire performance in different polymers, such as polyolefins [53] and polyurethanes [54]. In particular, previous studies have demonstrated synergistic effects in flame-retarded polylactide with different flame retardants, such as blends of APP/EG [18], with the thermal stability of FR/PLA blends over 300 °C. Moreover, it was also observed in the literature that the blends of EG and clays show an increase in the thermal stability of PLA at the same time, which improves the mechanical properties and reduces the flame propagation rate by the formation of a compact char [21]. In another study, Murariu describes how the addition of 6% of expanded graphite reduces the molecular weight of a PLA to around 50% [55], which is related to the presence of impurities. This affects mechanical properties with lower tensile strength, higher elastic modulus, and better fire performance compared to plain PLA (reduction of 30% pHRR with 12% filler). However, it is known that EGs could differ in properties depending on different parameters, such as the raw graphite used, the amount and nature of the intercalated species, and the production process. This renders a range of materials that differ in the set-up intumescence temperature and expansion rate. Different EG grades could address different polymer blend performances. In the present study, the nature of the used EG might have affected the stability of the polymer, leading to a loss of thermal stability.

Similar performance has been obtained for the PLA30IM formulation. In this case, the lower degradation temperature could be attributed to the early decomposition of the melamine-blowing agent and pentaerythritol, promoting the hydrolytic degradation of PLA. Similar results have been appreciated in PLA ATH blends, where the dehydration of ATH promotes the hydrolyzation of ester bonds in PLA. Instead, the addition of APP or CDP has little effect on the thermal stability of the polymer. Overall, a higher final residue than theoretical, regardless of whether they affect the stability of the polymer, proves the condensed phase action of more effective flame retardants: APP, EG, and IM. In these cases, the generation of polymer char promotes the reduction in heat release, as shown in Table 2.

The analysis of the molecular weight distribution of PLA in FR/PLA blends has been used to check if the incorporation of the flame retardant has a negative effect on the PLA matrix. Figure 4 shows the molecular weight distributions of formulations with flame retardants showing better fire performance (APP; EG; INT). The average molecular weight values in weight and number, Mw and Mn, respectively, and the polydispersity index were calculated (Table 4).

Since some studies have shown that the processing of PLA leads to a decrease in PLA molecular weight due to chain scission without affecting its chemical structure [56,57], the effect of sample manufacturing was first studied. In this case, the PLA molecular weight is hardly affected by processing conditions (Appendix A). However, the addition of different fire retardants seriously affects the molecular weight of the system (Figure 4). Thus, while the addition of APP leads to a decrease in Mw by almost half, the addition of EG drastically reduces this parameter; the higher the EG content, the greater the decrease. Similarly, the intumescent mixture produced a high decrease in the molecular weight, although to a lesser extent.

As far as the addition of EG is concerned, the thermogravimetric results had already pointed to the negative effect of EG on PLA since the initial degradation temperature decreased drastically when the EG content increased. The GPC characterization confirmed those results. Attending to the degradation mechanism, it can be considered that the depolymerization reaction is favored by the action of released acid species from EG during the material blending process [58]. However, in the case of APP, no destabilizing effect was observed by TGA, but a significant decrease in molecular weight was observed after processing. Again, the presence of acid species that catalyze hydrolytic degradation could probably explain this performance. The different mechanisms of PLA degradation can be attributed to the higher thermal stability of APP vs. EG. This suggests that the main mechanism of PLA degradation in this case is chain scission.

Considering the cone calorimeter test, TGA, and GPC analysis, APP appears to be the most suitable flame retardant for 3D printing grade FR-PLA production. Previous analysis has shown that it must be included by at least 30% since 15% was found to be insufficient regarding fire behavior. Taking these results into account, the thermo-mechanical properties of PLA30APP were studied by DMA (Figure 5). The storage modulus values around 3500 MPa at −75 °C and 2500 MPa at 25 °C observed for PLA30APP, as well as a glass transition temperature around 65 °C, confirmed its thermo-mechanical stability since these values are similar to those observed for the reference PLA. Moreover, in both cases, the recovery of the E′ modulus from 80 °C is due to the cold crystallization of the PLA.

As demonstrated by the DSC thermograms in Figure 6, the material remains in its amorphous state, and no significant nucleating effect of the APP is observed during the cooling of the material. The extrusion and/or compression processing of PLA results in an amorphous material with a very low crystallinity degree (see Table 5), which is able to crystallize once the glass transition is overcome. This behavior is also reflected by the tan δ height, which exceeds the value of 1 when the fully crystallized PLA barely reaches the value of 0.5 [59]. In conclusion, the incorporation of APP, despite significantly reducing the molecular weight of PLA, does not affect the thermal stability nor the mechanical or thermal characteristics of reference PLA.

### 3.3. Printability

Once the initial screening in terms of flame-retardant behavior is performed, the printability of the samples is studied. In order to analyze the effect of the incorporation of FR into the formulation, the rheological characterization of the samples was carried out. As shown in Figure 7A, the incorporation of the most suitable candidate, PLA30APP, resulted in a decrease in the viscosity of the material. This result is in good agreement with the polymer degradation behavior observed in GPC traces. In addition, at low shear rates, an increase in viscosity in the Newtonian plateau can be observed. This viscoplastic behavior can be attributed to the presence of ammonium polyphosphate particles within the polymer matrix. Several studies have reported decreases in material viscosity upon the addition of flame retardants, which is attributed to factors such as the disruption of polymer chain entanglements or the lubricating effect of the flame retardant particles within the polymer matrix [60]. Moreover, the observed viscoplastic behavior at low shear rates due to the presence of flame-retardant particles is consistent with previous studies examining polymer composites with particulate fillers. Similarly, the viscoelastic spectrum (Figure 7B) shows the alteration of the terminal zone, as the elastic modulus (G′) tends to stabilize at low frequencies. However, neither the material flow nor the interlayer adhesion was impeded by the presence of those particles. In fact, the material flow during the extrusion occurred in the high shearing regions (between 100 and 1000 s^−1^), and the material showed a predominantly liquid behavior, demonstrating the capability of polymer chains to diffuse in the formed interlayer. Similar investigations have shown that the incorporation of flame retardants can affect the rheological properties of the material but does not necessarily impede material flow or interlayer adhesion during the printing process. The capability of polymer chains to diffuse within the formed interlayer, as demonstrated in this study, has also been reported in the literature for various polymer systems with additives or fillers [61]. Thus, 3D-printed specimens were printed for both PLA and PLA30APP materials.

### 3.4. Comparison between 3D-Printed and Compression-Molded Samples

Finally, the influence of manufacturing on the final fire behavior and mechanical properties was investigated. Figure 8 shows the evolution of AHRE in time both for the compression-molded and 3D-printed specimens of PLA and PLA30APP. As shown, 3D-printed parts resulted in a slightly worse performance than the compression-molded counterparts. This effect might be related to the increase in the porosity of 3D-printed samples and/or additional degradation during manufacturing. Despite this, the incorporation of APP into the formulation increases the fire performance as the AHRE peak decreases for that sample. In addition, the objective of a reduction in MARHE values below 90 kW/m^2^ is still obtained (68.2 kW/m^2^ ± 1.2 for the PLA30APP (3D) sample).

Test specimens (for flexural, ISO 178 [37], and tensile, ISO 527 [38], properties) were produced both by injection and 3D printing processes for a comparative evaluation of the mechanical properties of samples produced by the two manufacturing methods. Thus, results obtained from the characterization of the injected and 3D-printed specimens are shown in the following tables. Table 6 shows the elongation at break (%), tensile strength (MPa), and Young’s modulus (MPa) values obtained by tensile tests.

The incorporation of the APP flame retardant into the PLA matrix increases the stiffness, reducing, at the same time, the toughness and flexibility of the material. The behavior is the same regardless of the process that is used for their fabrication (3D printing or injection process).

As shown in Table 7, test specimens in flexural characterization show the same behavior even though they are fabricated by 3D printing or injection processes. The incorporation of the APP flame retardant in the PLA matrix decreases the flexural strength (MPa), increasing the flexural modulus (MPa).

In all cases, test specimens fabricated by 3D printing show lower values than those fabricated by the injection process. This fact is always the case with any printed object, considering the meso-structural lapses [62].

## 4. Conclusions

In summary, our investigation into various flame-retardant combinations aimed at developing an FR/PLA grade suitable for additive manufacturing in the rail industry yields critical insights. The choice of flame retardant emerges as a pivotal factor influencing both the fire performance and stability of PLA compounds. Notably, flame retardants operating in the condensed phase, such as APP, EG, and IM, demonstrated superior fire performance compared to those acting in the gas phase, like CDP and ATH.

Our investigation further reveals that the PLA matrix exhibits heightened sensitivity to the decomposition products of flame retardants reacting at low temperatures. This sensitivity results in the release of substances that can induce the hydrolytic degradation of PLA ester bonds, with a pronounced impact observed in EG and ATH-based compounds. Among the tested flame retardants, APP stands out as the optimal candidate for the final application. A 30% dosage of APP in the formulation allows for the production of 3D-printed test specimens, meeting the stringent HL3 requirements specified in the EN45545-2 standard for certain applications within the railway sector.

This study underscores the significance of careful flame-retardant selection and dosage to achieve the desired fire performance and stability in 3D-printed PLA parts, particularly for applications with stringent safety standards, such as those in the railway industry. The identified formulations and their corresponding fire performance characteristics pave the way for advancements in flame-retardant PLA composites tailored for specific industrial requirements.

## Figures and Tables

**Figure 1 polymers-16-01030-f001:**
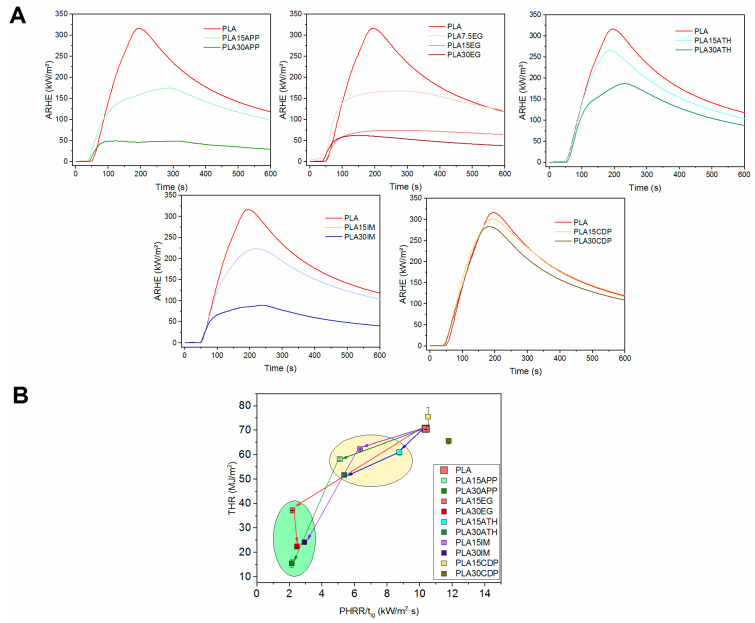
ARHE vs. time plots for different PLA/FR formulations (**A**) and comparative fire behavior through Petrella plots (**B**).

**Figure 2 polymers-16-01030-f002:**
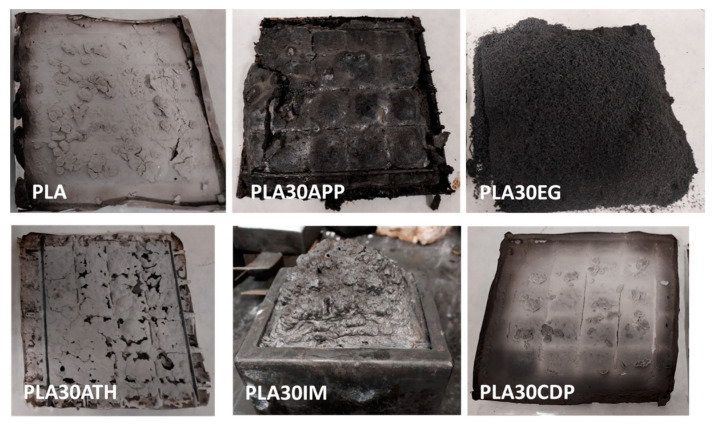
Obtained residues for cone calorimeter-tested samples with different flame retardants.

**Figure 3 polymers-16-01030-f003:**
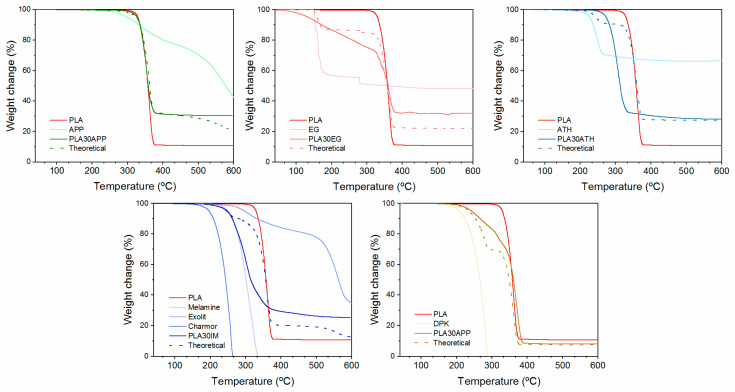
TGA of the selected formulations, PLA 30APP, PLA30EG, PLA30ATH, PLA30IM, and PLA30CDP.

**Figure 4 polymers-16-01030-f004:**
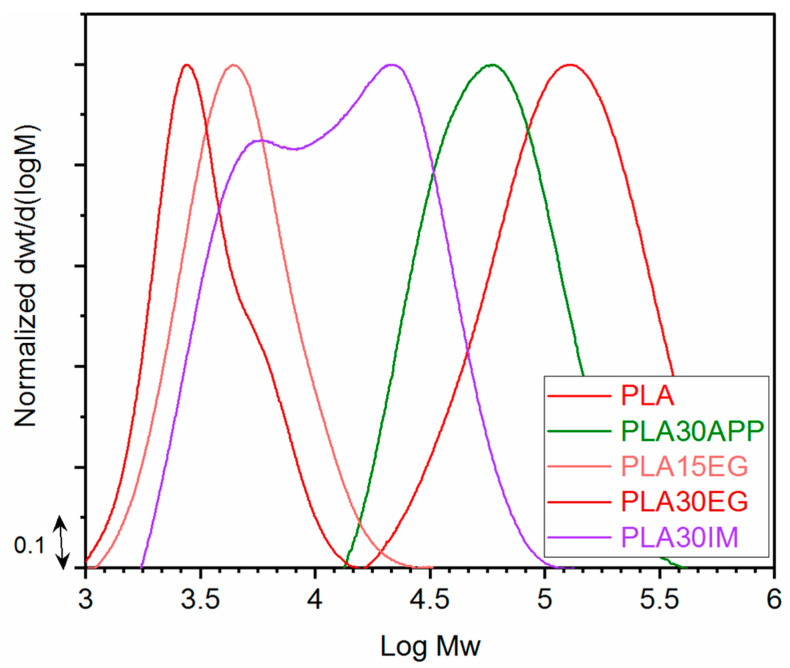
Molecular weight distributions and molecular parameters obtained by GPC for the selected formulations.

**Figure 5 polymers-16-01030-f005:**
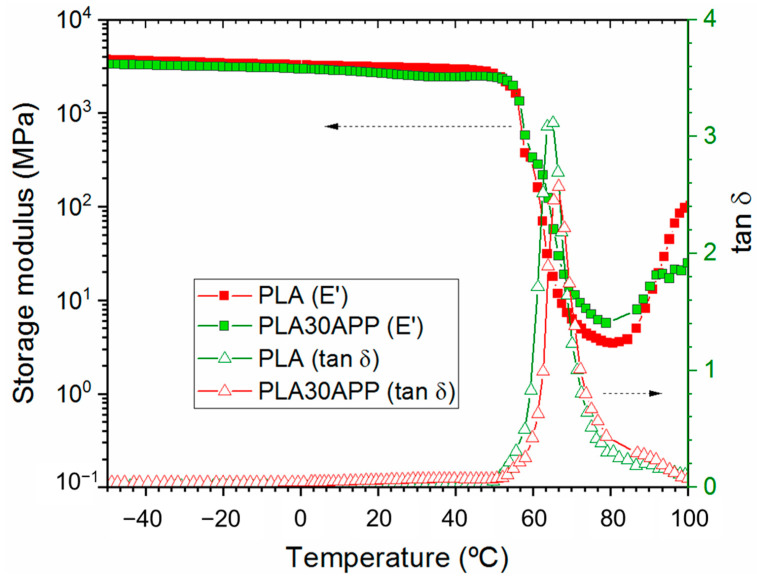
DMA thermogram for reference PLA (red symbols) and PLA 30 APP (green symbols).

**Figure 6 polymers-16-01030-f006:**
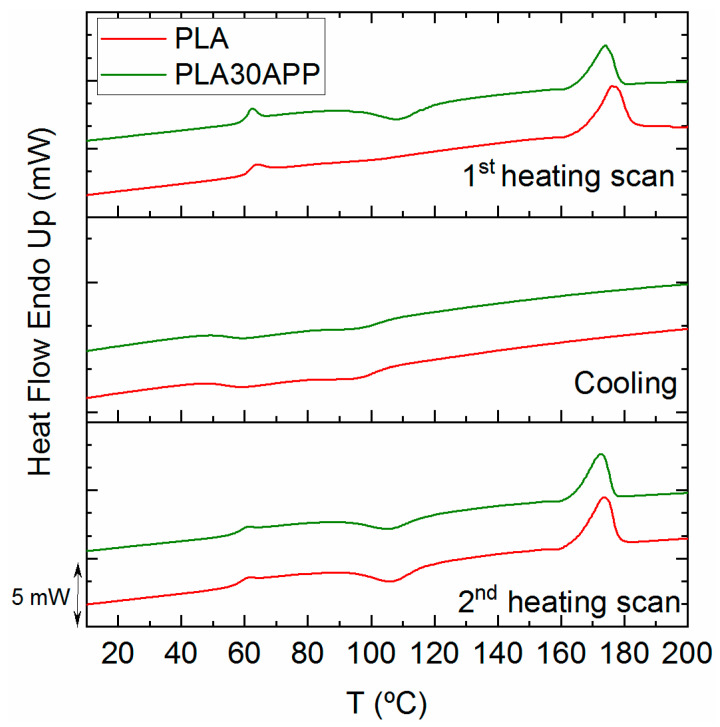
DSC traces for PLA and PLA30APP for the first heating scan (**top**), cooling (**middle**) and second heating scan (**bottom**).

**Figure 7 polymers-16-01030-f007:**
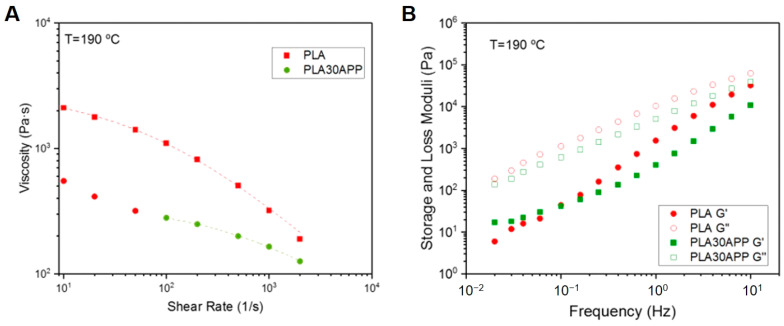
Viscosity curves (**A**) and viscoelastic spectra (**B**) at 190 °C for both PLA and PLA30APP samples. Representative 3D-printed parts using both materials.

**Figure 8 polymers-16-01030-f008:**
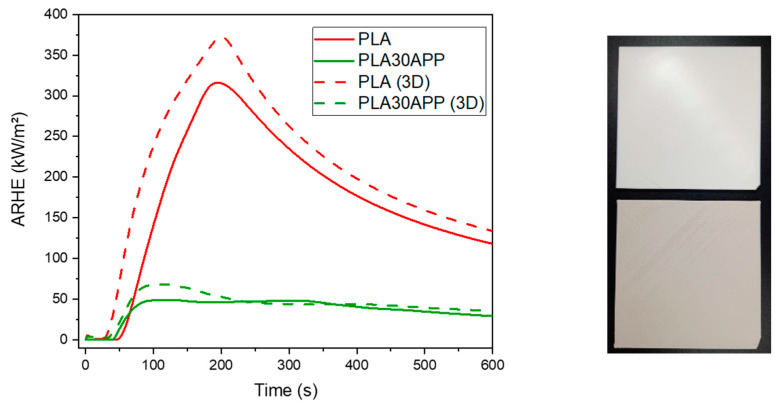
Comparison of fire behavior (ARHE vs. time plots) for 3D-printed (dotted line) and thermoconformed (solid line) PLA and PLA30APP specimens. The picture shows the test specimens produced by 3D printing, PLA (up) and PLA30APP (down).

**Table 1 polymers-16-01030-t001:** Compositions of the studied samples.

Sample	Flame Retardant Type	Composition (wt%)
PLA15APP	APP	15
PLA30APP	APP	30
PLA7.5EG	EG	7.5
PLA15EG	EG	15
PLA30EG	EG	30
PLA15ATH	ATH	15
PLA30ATH	ATH	30
PLA15IM	MEL/APP/PA	15 (3.75/7.5/3.75)
PLA30IM	MEL/APP/PA	30 (7.5/15/7.5)
PLA15CDP	CDP	15
PLA30CDP	CDP	30

**Table 2 polymers-16-01030-t002:** Results obtained for heat release and smoke generation.

Sample	t_ignition_ (s)	HRR_peak_ (kW/m^2^)	THR (MJ/m^2^)	MARHE (kW/m^2^)	CO/CO_2_ (×10^3^) Ratio	SEA (m^2^/kg)	Residue (%)
PLA	55.5 ± 1.5	575.7 ± 28.3	70.5 ± 0.1	310.8 ± 5.3	9.2 ± 0.8	3.3 ± 2.7	10.5 ± 0.9
PLA15APP	44.5 ± 0.5	226.7 ± 3.4	58.2 ± 0.5	174.5 ± 0.3	32.3 ± 1.5	12.6 ± 1.2	23.7 ± 0.3
PLA30APP	48.0 ± 0.1	102.8 ± 0.1	15.5 ± 1.5	49.3 ± 0.5	157.7 ± 42.2	56.6 ± 4.4	53.7 ± 0.2
PLA7.5EG	51.5 ± 0.2	306.9 ± 3.1	74.2 ± 1.2	165.0 ± 2.5	24.6 ± 0.8	6.2 ± 0.6	17.0 ± 1.5
PLA15EG	59.0 ± 1.0	129.3 ± 1.2	37.2 ± 0.1	73.5 ± 0.3	59.8 ± 2.5	2.5 ± 2.5	44.4 ± 0.1
PLA30EG	45.5 ± 0.5	112.3 ± 5.5	22.4 ± 0.3	61.1 ± 0.3	114.8 ± 1.9	3.7 ± 2.6	45.5 ± 0.7
PLA15ATH	53.0 ± 0.1	463.9 ± 10.0	61.1 ± 1.2	261.9 ± 4.0	18.3 ± 0.9	4.7 ± 0.2	19.9 ± 1.9
PLA30ATH	56.0 ± 3.1	300.8 ± 0.5	51.6 ± 0.6	190.7 ± 4.0	21.9 ± 1.1	1.8 ± 0.9	15.1 ± 0.6
PLA15IM	55.0 ± 0.2	348.9 ± 1.9	62.3 ± 0.2	226.7 ± 3.3	33.0 ± 4.6	19 ± 0.3	15.1 ± 0.6
PLA30IM	54.5 ± 0.5	159.1 ± 2.2	24.2 ± 0.5	91.3 ± 2.5	67.4 ± 7.7	5.8 ± 1.7	48.4 ± 1.4
PLA15CDP	52.0 ± 1.2	546.1 ± 22.1	75.5 ± 3.8	306.3 ± 5.6	76.5 ± 2.2	254.1 ± 23.5	9.0 ± 0.4
PLA30CDP	49.0 ± 1.1	577.0 ± 5.3	65.6 ± 1.2	288.6 ± 5.8	182.1 ± 1.5	485.8 ± 5.9	6.4 ± 0.6

**Table 3 polymers-16-01030-t003:** T_i_, T_max_ and residual mass values obtained from TGA and DTGA thermograms.

Sample	T_i_ (°C)	T_max_ (°C)	Residual Chart (%)
PLA	330	360	11
PLA30APP	327	350	30
PLA30EG	133	175/360	32
PLA30ATH	273	305	28
PLA30IM	248	300	25
PLA30CDP	242	318	8.2

**Table 4 polymers-16-01030-t004:** Molecular parameters obtained by GPC for the selected formulations.

Sample	M_n_ (g/mol)	M_w_ (g/mol)	PI
PLA	93,900	150,300	1.6
PLA30APP	48,300	70,200	1.4
PLA15EG	5600	9000	1.6
PLA30EG	3700	4800	1.3
PLA30IM	8300	17,000	2.0

**Table 5 polymers-16-01030-t005:** Thermal properties obtained by DSC for the final formulations.

Sample	1st Scan				Cooling	2nd Scan			
	T_g_ (°C)	T_cc_ (°C)	T_m_ (°C)	X_c_ (%)	T_cc_ (°C)	T_g_ (°C)	T_cc_ (°C)	T_m_ (°C)	X_c_ (%)
PLA	60	108	173	5	95	60	108	172	6
PLA30APP	57	102	176	12	95	57	105	173	5

**Table 6 polymers-16-01030-t006:** Mechanical properties of the injected and 3D-printed specimens obtained by tensile tests.

Sample	Young’s Modulus (MPa)	Tensile Strength (MPa)	Elongation at Break (%)
PLA (3D)	3664 ± 207	45.7 ± 1	3.5 ± 0.7
PLA30APP (3D)	5284 ± 1527	30.5 ± 2.4	2.0 ± 1.2
PLA	4584 ± 83	55.8 ± 1.7	5.7 ± 0.3
PLA30APP	5333 ± 63	49.3 ± 3	2.1 ± 0.7

**Table 7 polymers-16-01030-t007:** Mechanical properties of the injected and 3D-printed specimens obtained by flexural tests.

Sample	Flexural Modulus (MPa)	Flexural Strength (MPa)
PLA (3D)	2458 ± 98	55.83 ± 1.7
PLA30APP (3D)	3835 ± 54	49.26 ± 3.0
PLA	4399 ± 70	95.80 ± 3.6
PLA30APP	5209 ± 60	61.80 ± 0.8

## Data Availability

Data are contained within the article and Appendix A.

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
