# Peer review of "Development of Flame-Retardant Polylactic Acid Formulations for Additive Manufacturing"

_polymers, 2024, doi:10.3390/polym16081030_

Round 1

Reviewer 1 Report

Comments and Suggestions for Authors

The paper shows comprehensive research related to the investigation of the new material type. There are some issues, mostly related to the material selection and introduction:

1. I suggest using additive manufacturing instead of 3d printing.

2. PLA has a lot of disadvantages due to its weak resistance to UV, low glass transition temperature, etc. Please justify a selection of that type of material when there are a lot of different materials with better industry potential. 

3. An introduction needs to be significantly improved. There is a complete lack of a proper literature review that would constitute a proper background for the main topic. 

4. There is a lack of justification for the selection of the exact aim of this work. Please point out a potential usage of flame retardant PLA.

The whole part related to the research is made very well, but there is a lack of the results discussion with other available research results. Please fill those gaps. 

Author Response

REVIEWER 1

The paper shows comprehensive research related to the investigation of the new material type. There are some issues, mostly related to the material selection and introduction:

  1. I suggest using additive manufacturing instead of 3d printing.

Thank you for your valuable suggestion regarding the terminology used in our manuscript. We appreciate your attention to detail and your recommendation to differentiate between additive manufacturing and 3D printing within the context of our study.

We certainly take your suggestion into account and ensure that our revised manuscript accurately reflects the nuances between additive manufacturing as a broader term and the specific process of 3D printing. By making this distinction, we aim to provide clarity to our readers and contribute to the accurate portrayal of our research findings within the context of the broader field.

  1. PLA has a lot of disadvantages due to its weak resistance to UV, low glass transition temperature, etc. Please justify a selection of that type of material when there are a lot of different materials with better industry potential. 

Thank you for raising an important point regarding the selection of PLA as the material in our study. We appreciate your concern regarding its disadvantages, including its susceptibility to UV degradation and low glass transition temperature.

While it is true that PLA has certain limitations compared to other materials with potentially better industrial applications, it's important to understand the rationale behind our selection:

  • Environmental considerations: PLA is derived from renewable resources such as corn starch or sugarcane, making it a biodegradable and environmentally friendly option. This aligns with the growing emphasis on sustainability within various industries, including manufacturing.
  • Ease of processing: PLA is known for its ease of processing, making it an attractive choice for additive manufacturing processes such as 3D printing. Its relatively low melting temperature and good flow characteristics contribute to smoother printing and reduced energy consumption during processing.
  • Application specificity: While PLA may not be suitable for all industrial applications, it finds its niche in applications where its properties align well with the requirements. For instance, in applications where UV exposure is minimal or controlled, or where low glass transition temperature is not a limiting factor, PLA can offer a cost-effective solution.
  • Market acceptance and availability: PLA is widely accepted in the market and readily available, with established supply chains. This accessibility and familiarity make it a practical choice for certain applications, especially in industries where rapid prototyping or short production runs are common.

In light of these factors, our selection of PLA was made with careful consideration of its advantages and limitations, as well as its suitability for the specific requirements of our study. We acknowledge that other materials may offer certain advantages, but in this case, PLA best fulfilled our criteria and objectives. We trust that this explanation elucidates our reasoning behind choosing PLA as the material for our study, and we have expanded upon this justification in the introduction section of the revised manuscript.

  1. An introduction needs to be significantly improved. There is a complete lack of a proper literature review that would constitute a proper background for the main topic. 

We appreciate your feedback regarding the introduction of our manuscript. We acknowledge the need for a more comprehensive literature review to provide a robust background for the main topic. In our revised introduction, we have ensured to address this by thoroughly reviewing relevant literature and incorporating key findings and insights to establish a solid foundation for our study. This includes a critical analysis of prior research, highlighting gaps in knowledge, and identifying the specific contributions our study aims to make to the field. By enhancing the literature review section, we aim to provide readers with a clearer understanding of the context, significance, and novelty of our work within the broader scientific landscape. Thank you for bringing this matter to our attention. In response, we have diligently implemented the necessary improvements to our revised manuscript.

  1. There is a lack of justification for the selection of the exact aim of this work. Please point out a potential usage of flame retardant PLA.

Thank you for your insightful comment regarding the aim of our work. We recognize the importance of providing a clear justification for the specific objectives chosen for our study. One potential usage of flame retardant PLA lies in its application within the manufacturing of electronic enclosures and casings. In various industries, such as electronics, automotive and railway, there is a growing demand for materials that not only offer structural integrity but also possess flame retardant properties to enhance safety. Flame retardant PLA can serve as a viable solution in such scenarios, where its inherent biodegradability and mechanical properties, coupled with flame retardancy, make it suitable for producing protective housing for electronic components or vehicle parts. By focusing on the development and characterization of flame retardant PLA, our study aims to address the need for safer and more environmentally friendly materials in these high-demand applications. Thus, flame retardant PLA could be utilized in various components within railway systems, such as interior panels, seating, cable insulation, and housing for electronic equipment. We have ensured to incorporate this rationale and potential usage into the aim section of our manuscript to provide a clearer justification for our research objectives. Thank you for bringing this to our attention, and we have made the necessary adjustments accordingly.

The whole part related to the research is made very well, but there is a lack of the results discussion with other available research results. Please fill those gaps. 

Thank you for your feedback on the research section of our manuscript. We acknowledge the importance of discussing our results in relation to existing research findings to provide a more comprehensive understanding of the topic. In the revised version of our manuscript, we have expanded upon the results discussion by integrating comparisons with relevant studies in the field.

Reviewer 2 Report

Comments and Suggestions for Authors

In this study, to improve the fire performance of PLA ink for 3D printing in rail industry application, different flame retardants (FR) have been incorporated in the ink. Formulation with APP as the flame retardant was optimized to achieve a product with desired fire resistance and mechanical properties.

The manuscript is valuable and written well. However, authors should address the issues.

·         Authors should explain the structure and physiochemical properties of PLA polymer like melting point, in “Introduction section”. The paragraph can be enriched with literature reports about hot melt extrusion of PLA polymers. Other applications of 3D printing of PLA for example in drug delivery field can be mention (https://doi.org/10.1016/j.ijpharm.2023.123366)

·         Please mention the MW of the applied PLA polymer.

·         Authors should explain how they selected the composition percentage of the materials mentioned in Table1.

Best

Comments on the Quality of English Language

Minor English editing is needed.

Author Response

REVIEWER 2

In this study, to improve the fire performance of PLA ink for 3D printing in rail industry application, different flame retardants (FR) have been incorporated in the ink. Formulation with APP as the flame retardant was optimized to achieve a product with desired fire resistance and mechanical properties.

Thank you for your valuable feedback. We appreciate your positive assessment of our manuscript. We will address the issues you raised to enhance the clarity and completeness of the study.

The manuscript is valuable and written well. However, authors should address the issues.

  • Authors should explain the structure and physiochemical properties of PLA polymer like melting point, in “Introduction section”. The paragraph can be enriched with literature reports about hot melt extrusion of PLA polymers. Other applications of 3D printing of PLA for example in drug delivery field can be mention (https://doi.org/10.1016/j.ijpharm.2023.123366)

Thank you for your insightful suggestion regarding the enhancement of the Introduction section. In response to your feedback, we have enriched the paragraph discussing the structure and physiochemical properties of PLA polymer by including details such as its melting point. Additionally, we have incorporated literature reports about hot melt extrusion of PLA polymers to provide a more comprehensive understanding of this manufacturing process. Furthermore, we have expanded upon the discussion of other applications of 3D printing of PLA, including its utilization in the field of drug delivery, as exemplified by the study referenced (https://doi.org/10.1016/j.ijpharm.2023.123366). These additions will offer readers a broader perspective on the versatility and potential of PLA in various industries, further enriching the Introduction section of our manuscript. Thank you for your valuable input, and we have ensured to incorporate these enhancements into the revised version of our work.

  • Please mention the MW of the applied PLA polymer.

Thank you for your request to include the molecular weight (MW) of the applied PLA polymer. In the revised manuscript, we have ensured to provide this crucial information, offering transparency and clarity regarding the characteristics of the PLA used in our study. This addition will allow readers to better understand the properties and behavior of the polymer under investigation. We appreciate your attention to detail, and we have incorporated the MW (1503000 g·mol-1) of the PLA polymer in the materials and methods section of our manuscript.

  • Authors should explain how they selected the composition percentage of the materials mentioned in Table1.

The composition percentage of the materials mentioned in Table 1 was selected based on a combination of factors including compatibility with PLA, desired fire resistance properties, and consideration of mechanical performance. We have provided a more detailed explanation of the rationale behind the selection of composition percentages in the materials and methods section in the revised manuscript.

Thank you for your valuable suggestions. We have incorporated these improvements into the manuscript to ensure a more comprehensive and informative presentation of our research.

Round 2

Reviewer 1 Report

Comments and Suggestions for Authors

All my comments have been well addressed  the paper can be published

Author Response

We are sincerely grateful to you for your thorough and insightful review of the manuscript, which has significantly contributed to its enhancement and scholarly quality.